

# Factors associated with work engagement of nurses in the radiology department, Japan: a cross-sectional study

Hitomi Tsuchihashi[1,2], Takumi Yamaguchi[1,3,4,5], Yumiko Yamada[6], Tamami Koyama[7] and Yuko Matsunari[1]

[1] School of Health Sciences, Kagoshima University, Kagoshima, Kagoshima, Japan
[2] Department of Nursing, Kagoshima University Hospital, Kagoshima, Kagoshima, Japan
[3] School of Nursing, Tokyo Medical University, Shijuku, Tokyo, Japan
[4] Department of Biostatistics, Saitama Medical University, Iruma, Saitama, Japan
[5] Nuclear Safety Research Association, Minato, Tokyo, Japan
[6] School of Nursing, Kwassui Women's University, Omura, Nagasaki, Japan
[7] School of Nursing, Tsuruga Nursing University, Tsuruga, Fukui, Japan

Corresponding author
Takumi Yamaguchi,
yamaguchi.takumi.8t@tokyo-med.ac.jp

## ABSTRACT

**Background:** Radiology departments present unique challenges compared to other departments due to exposure to radiation and the specialized nature of the work. Nurses must not only manage typical nursing duties but also adhere to strict safety protocols to minimize radiation exposure. These additional responsibilities can significantly impact their work engagement and overall job satisfaction.

**Objective:** This study aimed to identify the factors associated with work engagement among nurses working at prefectural designated cancer care hospitals in Japan. Identifying these factors may lead to improvements in future work environments and educational systems.

**Methods:** This was a cross-sectional study using an internet-based survey. A questionnaire using the Utrecht Work Engagement Scale (UWES) was conducted among 317 nurses; 140 responded (response rate: 44.2%).

**Results:** Significant associations were found between work engagement and several factors. The mean UWES score was 54.3 (Standard deviation (SD): 18.4). Work engagement was positively associated with age (B = 0.179, $p$ = 0.03), being male (B = 0.19, $p$ = 0.015), higher position (B = 0.199, $p$ = 0.012), desire for radiology assignments (B = 0.223, $p$ = 0.003), and presence of a radiation exposure consultation system (B = 0.214, $p$ = 0.034).

**Conclusions:** This study identified several factors associated with work engagement among radiology nurses, highlighting the importance of specialized support systems to address radiation-related concerns. These findings can inform interventions to enhance work engagement and well-being in this field.

## INTRODUCTION

In recent years, work engagement surveys of nurses have been actively conducted, highlighting the importance of job satisfaction in the nursing profession

(*da Silva et al., 2020*; *Othman & Nasurdin, 2019*). Work engagement describes a condition of work-related well-being that is the opposite of burnout (*Bakker et al., 2008*), involving positive and fulfilling emotions and cognitions. Understanding work engagement at its core is essential for a comprehensive analysis. According to *Ferraro et al. (2020)*, work engagement is positively related to work performance, organizational effectiveness, and workers' well-being, measured using the Utrecht Work Engagement Scale (UWES). Conversely, low levels of work engagement are associated with increased rates of burnout.

Previous studies of work engagement among nurses using the UWES have found significant positive correlations between work engagement and working environment (*Wan et al., 2018*), positive personality, competencies, and job performance (*Hu et al., 2021*). Thus, the foundation of quality achievement goals affects work engagement. In addition, a study comparing work engagement among nurses and nurse managers reported that nurses had lower work engagement and commitment to the organization than nurse managers (*Al-Dossary, 2022*).

The International Council of Nurses' report on occupational health and safety for nurses noted that the working environment of healthcare workers is considered to be one of the most hazardous professional environments, and that nurses are often exposed to health hazards (*International Coucil of Nurses, 2024*). Healthcare hazards include biological hazards (*e.g.*, viruses and bacteria), chemical hazards (*e.g.*, glutaraldehyde and cytotoxic drugs), ergonomic hazards (*e.g.*, overexertion, falls, lifting), physical hazards (*e.g.*, radiation, sharp objects), and psychological hazards (*e.g.*, shift work, excessive workload, violence, and stress). In Japan, the number of radiology nurses has doubled from 58,827 in 2002 to 114,670 in 2020. In addition, the number of cancer cases according to regional cancer registries is increasing, and it is predicted that patients will have more opportunities to receive radiological treatment. *Fujibuchi et al. (2021)* reported that 1.17% of nurses had lens-equivalent doses exceeding 20 mSv, and those involved in outpatient care and endoscopy tended to have higher doses.

Radiology departments present unique challenges compared to other departments due to exposure to radiation and the specialized nature of the work. Nurses in these departments must not only manage typical nursing duties but also adhere to strict safety protocols to minimize radiation exposure. These additional responsibilities can significantly impact their work engagement and overall job satisfaction. Internationally, the roles and education of professionals in radiology services vary, with radiographers in Europe and technologists in the United States having different scopes of practice and educational requirements compared to nurses in Japan. This diversity in roles can influence the way work engagement is perceived and managed across different countries.

Nurses in Japan are increasingly involved in radiological treatment. However, in a survey of nurses working in medium-sized hospitals, 85.9% of nurses responded that they would not like to work in a radiology department (*Nagatomi et al., 2019*). In addition, in a survey of nurses engaged in interventional radiology regarding their perceptions of occupational exposure, more than 70% of respondents indicated that they were concerned about the health effects of radiation from engaging in interventional radiology

(*Masujima & Noto, 2018*). In addition, nurses were found to be concerned about uncertain exposure doses and effects (*Oishi et al., 2018*).

Therefore, the present study aimed to identify the factors associated with work engagement among nurses working in radiology. The results of the present study may help to improve the work engagement of nurses working in radiology departments.

## MATERIALS AND METHODS

### Study design
Cross-sectional study.

### Setting
This study was conducted in prefectural designated cancer care hospitals across Japan. The survey was carried out from December 2021 to February 2022 using an internet-based questionnaire.

### Participants
The target population was nurses working in radiology departments of prefectural designated cancer care hospitals in Japan. We employed a convenience sampling method. First, consent documents for participation were sent to all 51 prefectural designated cancer care hospitals in Japan, of which 24 hospitals agreed to participate. We then distributed descriptive documents to all nurses working in the radiology departments of these 24 hospitals ($N = 317$). Subsequently, 140 participants responded (response rate: 44.2%) (Fig. 1).

The sample size for this study was determined *post-hoc*. Given the exploratory nature of our research, the goal was to identify a wide range of relationships and patterns using the full sample at our disposal. Consequently, we deemed the number of samples collected as appropriate for the objectives of our study.

### Tools of data collection
1. Demographic questionnaire: We collected information on age, gender, marital status, presence of children under 18, years of nursing experience, years in radiology, education level, position, qualifications, desire for radiology assignment, and principal operations.

2. Utrecht Work Engagement Scale (UWES): The Utrecht Work Engagement Scale (UWES) was developed by Shaufele and Baller and is a widely used scale for measuring work engagement comprising three dimensions: "vitality", "dedication", and "absorption" in work (*Schaufeli & Bakker, 2004*). In this study, the Japanese version of the UWES utilized has undergone linguistic validation in 2008 by *Shimazu et al. (2008)* demonstrating reliable and valid results in terms of its internal consistency (Cronbach's alpha = 0.92) and test–retest reliability over a 2-month interval (0.66).

3. Perceptions of radiation control: Six items were set regarding the perceptions of radiation control and radiation doses, assessed on a 7-point Likert scale ranging from "very much disagree" (0) to "very much agree" (6).

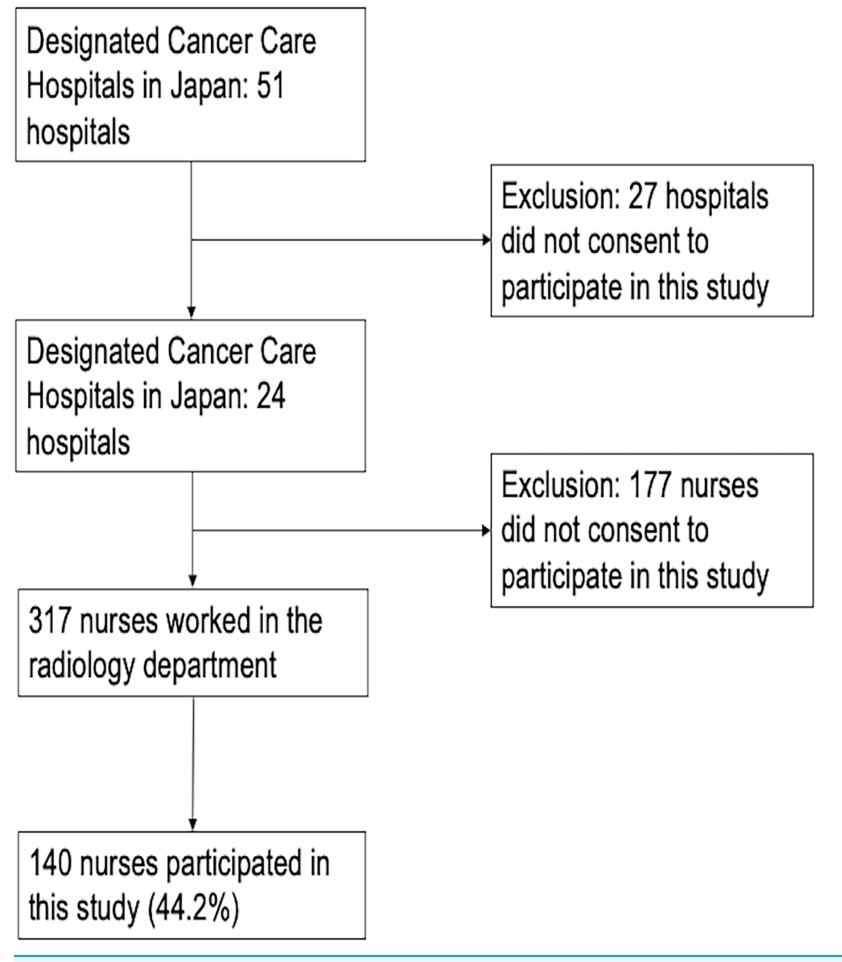

**Figure 1 Enrollment of participants in this study.**

## Statistical methods

First, we calculated descriptive statistics for all the items. In addition, we used single regression analysis and selected variables with $p < 0.1$ as explanatory variables (*Shao, 1996*). Multiple regression analysis was performed using UWES scores as the outcome variable. The forced entry method was employed for model selection. All variance inflation factors in the multiple regression analysis were ≤1.969, with values of less than 2.5 indicating that multicollinearity is not a concern. All data were statistically analyzed using SPSS version 28 (IBM, Tokyo, Japan). Statistical significance was set at $p < 0.05$.

## Ethical consideration

After providing all the necessary information about the study and obtaining informed consent, participants were asked to complete the questionnaire. Administrators of the participating facilities and the subjects themselves were informed in writing about the purpose of the survey and that participation was entirely voluntary. Consent for participation in the study was inferred from the return of the completed questionnaire. The present study was conducted in accordance with the tenets of the Declaration of

Helsinki and approved by the Ethics Committee of Kagoshima University Epidemiological Study (approval number: 210184) on November 17, 2021.

## RESULTS

### Descriptive data

Table 1 presents the sociodemographic characteristics of the 140 study participants. The majority of participants were in their 40s (38.5%) or 50s (28.6%). Women comprised the vast majority of the sample (92.9%, $n = 130$). Most nurses (85.0%, $n = 119$) had more than 11 years of nursing experience. However, when it came to experience specifically in radiology departments, nearly half (48.6%, $n = 68$) had worked there for 4 years or less, indicating that many experienced nurses were relatively new to radiology. Regarding professional roles, 21.4% ($n = 30$) of participants held nurse manager positions. In terms of personal life, 68.6% ($n = 96$) were married, and 55.7% ($n = 78$) had minor children. In our study, 25.7% ($n = 36$) of the participating nurses had acquired post-employment qualifications in areas related to nursing services, such as certified nursing, interventional nursing expertise, gastrointestinal endoscopy, as well as in other fields like aromatherapy and medical information technology. Interestingly, only 22.9% ($n = 32$) expressed a desire to be assigned to the radiology department. This suggests that many nurses working in radiology may not have initially chosen this specialization. The most common type of work among participants was angiography, performed by 60.0% ($n = 84$) of the nurses. This was followed by radiotherapy (47.9%, $n = 67$), endoscopy/fluoroscopy (42.1%, $n = 59$), nuclear medicine (40.7%, $n = 57$), and diagnostic imaging (38.6%, $n = 54$). It's worth noting that many nurses worked in multiple areas within radiology.

### Perceptions of radiation protection and health effects

Table 2 presents the participants' perceptions regarding radiation protection and radiation health effects. The responses were measured on a scale from 0 to 6, with higher scores indicating stronger agreement with the statement. Nurses generally felt that radiation exposure management was being carried out appropriately at their workplace (mean = 4.2, standard deviation (SD) = 1.5) and that their own radiation protection measures were sufficient (mean = 4.1, SD = 1.3). However, there was less certainty about the availability of consultation systems for radiation exposure concerns (mean = 3.7, SD = 1.6). Interestingly, participants expressed relatively low concern about their personal dosimeter readings (mean = 2.7, SD = 1.5). This could suggest either low exposure levels or a need for more education about interpreting dosimeter data. Regarding health effects, nurses showed moderate concern about the impact of occupational radiation exposure on their own health (mean = 2.9, SD = 1.5). However, they expressed less concern about potential effects on their children's health (mean = 2.1, SD = 1.7), which was the lowest scored item in the survey. These results indicate that while nurses generally feel protected and well-managed in terms of radiation exposure at work, there may be room for improvement in consultation systems and education about long-term health effects of radiation exposure.

**Table 1 Participants' sociodemographic factors.**

| Variables | Options | n = 140<br>n (%) |
|---|---|---|
| Age | 20–29 years old | 11 (7.9) |
| | 30–39 years old | 29 (20.7) |
| | 40–49 years old | 54 (38.5) |
| | 50–59 years old | 40 (28.6) |
| | 60 years old or older | 6 (4.3) |
| Gender | Female | 130 (92.9) |
| | Male | 10 (7.1) |
| Years as nurse | 0–4 years | 8 (5.7) |
| | 5–10 years | 13 (9.3) |
| | 11 years or more | 119 (85.0) |
| Nursing experience with radiology department (years) | 0–4 years | 68 (48.6) |
| | 5–10 years | 53 (37.9) |
| | 11 years or more | 19 (13.6) |
| Position | Staff nurse | 110 (78.6) |
| | Nurse manager | 30 (21.4) |
| Marital Status | Married | 96 (68.6) |
| | Unmarried | 44 (31.4) |
| Having minor children | Yes | 78 (55.7) |
| | No | 62 (44.3) |
| Acquisition of qualifications after employment | Yes | 36 (25.7) |
| | No | 104 (74.3) |
| Desire to be assigned to the radiology department | Desired | 32 (22.9) |
| | Not desired | 108 (77.1) |
| Place of work | Diagnostic imaging | 54 (38.6) |
| | Angiography | 84 (60.0) |
| | Endoscopy/fluoroscopy | 59 (42.1) |
| | Nuclear medicine | 57 (40.7) |
| | Radiotherapy | 67 (47.9) |

**Note:**
UWES, utrecht work engagement scale. For 'Place of work', participants could select multiple options.

## Work engagement scores

Table 3 presents the Utrecht Work Engagement Scale (UWES) scores of the participants. The mean (SD) total UWES score was 54.3 (18.4). For the subscales, the mean (SD) scores were as follows: vigor 17.8 (7.0), dedication 18.0 (6.2), and absorption 18.6 (6.8). These scores suggest a moderate level of work engagement among the participants, with slightly higher scores in the absorption dimension.

## Factors associated with work engagement

Table 4 shows the results of the multiple regression analysis, which identified several factors significantly associated with UWES scores:

**Table 2 Nurses' perceptions of radiological protection and radiation health effects.**

| | $n = 140$ |
|---|---|
| **Variables** | **Mean (SD)** |
| How do you feel about the dose on your personal dosimeter? | 2.7 (1.5) |
| Is radiation exposure management being carried out appropriately at your workplace? | 4.2 (1.5) |
| Does your workplace have a system in place for consultation about exposure to radiation? | 3.7 (1.6) |
| Do you think your radiation protection measures are sufficient? | 4.1 (1.3) |
| Do you think there is an effect on your health from occupational radiation exposure? | 2.9 (1.5) |
| Do you think your radiation exposure will affect the health of your children? | 2.1 (1.7) |

Note:
Responses were measured on a scale from 0 to 6, with higher scores indicating stronger agreement. SD, standard deviation.

**Table 3 UWES scores.**

| | $n = 140$ |
|---|---|
| **Variables** | **Mean (SD)** |
| Overall score | 54.3 (18.4) |
| Vigor | 17.8 (7.0) |
| Dedication | 18.0 (6.2) |
| Absorption | 18.6 (6.8) |

Note:
SD, standard deviation.

1. Age (B = 0.179, t = 2.192, $p$ = 0.03): Older nurses demonstrated higher engagement.

2. Gender (B = 0.19, t = 2.462, $p$ = 0.015): Male nurses showed higher engagement compared to female nurses.

3. Position (B = 0.199, t = 2.547, $p$ = 0.012): Nurse administrators exhibited higher engagement than staff nurses.

4. Desire to be assigned to a radiology department (B = 0.223, t = 2.992, $p$ = 0.003): Nurses who expressed a desire to work in radiology showed higher engagement.

5. Having a consultation system for radiation exposure in the workplace (B = 0.214, t = 2.146, $p$ = 0.034): The presence of such a system was associated with higher engagement scores.

Among these factors, the desire to be assigned to a radiology department showed the strongest association with work engagement (highest B value), followed by the presence of a consultation system for radiation exposure. It's noteworthy that while only 22.9% of nurses expressed a desire to be assigned to the radiology department (as reported in Table 1), this factor had the strongest association with work engagement.

Interestingly, several factors that showed significant associations in the single regression analysis did not remain significant in the multiple regression model. These included the number of years working as a nurse, marital status, and post-employment qualifications.

**Table 4 Factors associated with UWES were assessed using single/multiple regression analysis.**

| Variables | Single regression analysis | | Multiple regression analysis | | | |
|---|---|---|---|---|---|---|
| | B | p-value | B | t | p-value | VIF |
| Age | 0.347 | <0.001 | 0.179 | 2.192 | 0.03 | 1.298 |
| Gender (reference: female) | 0.225 | 0.008 | 0.19 | 2.462 | 0.015 | 1.172 |
| Number of years working as a nurse | 0.233 | 0.006 | | | | |
| Nursing experience in radiology department (years) | 0.215 | 0.011 | 0.101 | 1.275 | 0.223 | 1.318 |
| Marital status (reference: single) | 0.197 | 0.02 | 0.131 | 1.677 | 0.096 | 1.203 |
| Having young children (reference: not having) | 0.02 | 0.813 | | | | |
| Position (reference: staff) | 0.215 | 0.011 | 0.199 | 2.547 | 0.012 | 1.2 |
| Post-employment qualifications (reference: none) | 0.203 | 0.016 | 0.045 | 0.567 | 0.572 | 1.247 |
| Desire to be assigned to the radiology department (reference: no) | 0.306 | <0.001 | 0.223 | 2.992 | 0.003 | 1.093 |
| How do you feel about the dose on your personal dosimeter? | 0.031 | 0.717 | | | | |
| Is radiation exposure management being carried out appropriately at your workplace? | 0.264 | 0.002 | 0.08 | 0.798 | 0.427 | 1.969 |
| Does your workplace have a system in place for consultation about exposure to radiation? | 0.306 | <0.001 | 0.214 | 2.146 | 0.034 | 1.939 |
| Do you think your radiation protection measures are sufficient? | 0.157 | 0.065 | −0.022 | −0.234 | 0.815 | 1.695 |
| Do you think there is an effect on your health from occupational radiation exposure? | −0.159 | 0.061 | −0.008 | −0.11 | 0.913 | 1.16 |
| Do you think your radiation exposure will affect the health of your children? | −0.015 | 0.861 | | | | |
| Diagnostic imaging | 0.047 | 0.581 | | | | |
| Angiography | −0.049 | 0.562 | | | | |
| Endoscopy/fluoroscopy | 0.049 | 0.565 | | | | |
| Nuclear medicine | 0.154 | 0.07 | 0.114 | 1.555 | 0.122 | 1.049 |
| Radiotherapy | 0.12 | 0.159 | | | | |
| R | | | 0.597 | | | |
| $R^2$ | | | 0.356 | | | |
| Adjusted $R^2$ | | | 0.295 | | | |

Note:
VIF, variance inflation factors.

This suggests that these factors may be interrelated or their effects may be mediated by other variables in the model.

The importance of having a consultation system for radiation exposure aligns with the perceptions reported in Table 2, where nurses showed moderate uncertainty about the availability of such systems (mean = 3.7, SD = 1.6). This finding underscores the potential impact of organizational support on work engagement in high-risk environments like radiology departments.

These findings suggest that both individual characteristics (age, gender, position, personal preference) and workplace factors (consultation system) play significant roles in determining work engagement among nurses in radiology departments. The results highlight the complex interplay of personal and organizational factors in shaping work engagement in this specialized nursing field.

## DISCUSSION

We conducted a survey of radiology nurses at prefectural designated cancer care hospitals in Japan using the UWES to identify factors related to work engagement. In the 2021 survey of the nursing workforce in Japan, the most prevalent age group was 40–49 years, accounting for 28.9%, and 93.5% of the nurses were female, as reported in *Japan Nursing Association (2022)*. In a study conducted on nurses involved in radiological services in Japan, the predominant age group was also in their forties, constituting 36.6% of the sample, with women making up 93.1% of this group (*Oishi et al., 2018*). The age and gender demographics of the participants in the current study were found to be similar to those of Japanese nurses in general and those working in radiological services.

### UWES scores of radiology nurses and comparison with previous studies

A comparison of the UWES scores in this study with previous reports of UWES for nurses in Japan showed that nurses in hospitals working in the Tokyo metropolitan area scored 35.9 (9.8) (*Matsuoka & Tanaka, 2022*), while those in hospitals in the Kanto and Kansai areas scored 32.5 (9.0), indicating that the UWES scores in this study were higher (*Ishitsuka & Miki, 2016*; *Suto & Ishii, 2017*). These references to previous literature indicate that the subjects were more than 10 years younger than those in the present study, and it is assumed that the UWES scores in the present study were higher because it has been reported that age is associated with work engagement. We also found that participants' UWES scores in this study were lower than those reported by nurses in Brazil (*da Silva et al., 2020*), Saudi Arabia (*Aboshaiqah et al., 2016*), Iran (*Torabinia et al., 2017*), and China (*Wan et al., 2018*). Previous reports have shown that national characteristics and other factors also affect UWES scores, and Japanese people have been reported to have relatively low work engagement scores (*Matsuoka & Tanaka, 2022*; *Shimazu et al., 2010*). The results of the current study are consistent with these previous findings. While our study aligns with previous findings on the influence of age and gender on work engagement, it uniquely highlights the importance of workplace systems for radiation exposure consultation in enhancing work engagement among radiology nurses. This suggests a more nuanced understanding of work engagement that considers not just individual but also organizational factors. In contemplating the international differences in UWES scores observed in this study, it is evident that the scores of nurses in Japan are lower compared to other countries. These disparities may be attributed to varying work environments, cultural backgrounds, and societal perceptions of the nursing profession in different countries. For instance, in Japan, the stringent work environment and prevalent long working hours may contribute to lower levels of work engagement among nurses. Additionally, the collectivist culture in Japan might impose limitations on individual self-expression and autonomy in career choices, potentially suppressing work engagement. Conversely, in countries like Brazil and Saudi Arabia, the social perception of the nursing profession and workplace support systems might differ, positively influencing the enthusiasm and engagement of nurses in their duties. Particularly in these countries, the

nursing profession may be more highly valued, and the presence of robust support and educational opportunities within the workplace could be factors enhancing job engagement.

### Age, gender, position, desire to be assigned to the radiology department, and having a consultation system for radiation exposure in the workplace were significantly related to UWES scores of radiology nurses

Previous studies of age and work engagement reported that work engagement increases with age (*Nakamura & Yoshioka, 2016*; *Obata & Morishita, 2014*; *Ozawa, Sugaya & Mori, 2022*), in accord with the findings of the current study. In addition, previous reports on the relationship between gender and work engagement have shown that men are more engaged than women, which is consistent with the results of this study (*Ogiso & Itoh, 2019*; *Sato et al., 2021*; *Schaufeli & Bakker, 2004*). Furthermore, higher staff positions have been reported to be associated with work engagement, which is consistent with the results of this study (*Al-Dossary, 2022*; *Nakamura & Yoshioka, 2016*; *Ogiso & Itoh, 2019*; *Sakurama, Yamada & Nakajima, 2021*). Furthermore, previous studies have shown that proactive personality influences UWES scores (*Hu et al., 2021*). In the current study, nurses who wished to be assigned to the radiology department tended to have higher UWES scores. Thus, it is likely that their desire for assignment would be fulfilled, and the results suggest that they would be enthusiastic about their work. Therefore, it is assumed that the results of this study have some validity, on the basis of consistency of the study results regarding demographic factors.

Previous studies have reported that factors related to UWES scores are associated with supervisor and organizational support (*Jasiński & Derbis, 2023*; *Kiema-Junes et al., 2020*), with results similar to those of the present study. Previous studies that have identified reasons for reluctance to work in radiology have reported that nurses are overly fearful of radiation because of a lack of knowledge and are unable to answer questions about the health effects of radiation on their patients (*Nagatomi et al., 2019*; *Yamada et al., 2019*). On the other hand, many nurses wanted to learn more about radiation, suggesting the need for a radiation exposure consultation system. In Japan, there are approximately 400 registered certified nurses in radiation oncology nursing, who have been working as experts in the field of radiation oncology nursing (*Japanese Nursing Association, 2024b*). As experts in cancer radiation therapy, they possess the ability to support patient treatment and provide consultation to staff involved in associated nursing care. In February 2022, the Japan Nurses Association approved Certified Nurse Specialists (CNS) in radiological nursing, and in November of the same year, three nurses began working as CNSs (*The Radiological Nursing Society of Japan, 2023*).

The role of CNSs in radiological nursing is to establish a system to deal with peacetime radiation accidents and disasters, and to provide nursing care to people with health problems associated with the field of radiological nursing (*Japanese Nursing Association, 2024a*). The current results indicate that the establishment of a consultation system for occupational exposure is associated with better work engagement of nurses working in

radiology departments. We believed that radiological nursing specialists, due to the nature of their field, possess knowledge in strategies to reduce occupational exposure and in risk communication. The participants of this study were engaged not only in cancer radiation therapy but also in nursing for radioisotope therapy, intravascular radiology, endoscopic examinations, and other tests, necessitating nursing and exposure protection measures tailored to the characteristics of each examination room. We considered that the intervention of CNSs can contribute to the establishment of a consultation system for occupational exposure in the workplace. Thus, our findings suggest that the deployment of nurses with more knowledge than certified nurses (*i.e.*, CNSs in radiological nursing) in many hospitals, particularly prefectural designated cancer care hospitals, may help to reduce anxiety regarding radiation exposure among nurses working in radiology departments. Our study underscores the potential impact of CNS roles in radiological nursing on work engagement. We posit that the introduction of CNS roles could serve as a catalyst for improving work engagement by providing a structured system for addressing radiation-related anxieties among nurses. In the context of international nursing scenarios, the study on the education program for young nurses has shown that young nurses who received longer preceptorship became more involved in the nursing organization and increased their self-efficacy through the leadership function of preceptors (*Choi & Yu, 2022*). Previous surveys have clarified that organizational support and self-efficacy are positively correlated with work engagement (*Al-Hamdan & Bani Issa, 2022*). It is also conjectured that self-efficacy, enhanced through preceptorship, leads to improvement in work engagement. For nurses working in the radiology department, it is expected that the work engagement will be further enhanced with the support of highly specialized nurses in radiology. In light of our findings and existing literature, we argue that extending preceptorship in international nursing scenarios could be a strategic move to enhance work engagement. Such preceptorships could offer a more comprehensive support system, thereby increasing self-efficacy and engagement among radiology nurses.

It is crucial to focus on how the individual factors identified in our study impact the work engagement of radiology nurses. The factors such as age, gender, position, desire to be assigned to the radiology department, and the presence of a consultation system for radiation exposure in the workplace have been shown to be significant elements in enhancing nurses' work engagement. Understanding how these factors contribute to an individual nurse's enthusiasm for work and job satisfaction is essential for improving the quality and effectiveness of work in radiological nursing. These insights highlight the complexity of factors influencing work engagement in radiological nursing and suggest that interventions to enhance engagement should be multifaceted, addressing not just individual desires and skills, but also broader organizational and cultural aspects.

## Strengths and limitations

A major strength of the current study is that it is the first to identify factors related to work engagement among radiology nurses. In this study, the results of the multiple regression analysis revealed an adjusted $R^2$ of 0.295, which is not high. This suggests the presence of variables related to UWES that we did not anticipate, and not being able to account for

these variables can be considered a limitation of our study. In a previous research study, *Othman & Nasurdin (2019)* reported an adjusted $R^2$ of 0.15, while *Sakurama, Yamada & Nakajima (2021)* found an adjusted $R^2$ of 0.493. The variation in $R^2$ results in similar studies indicates that a range of factors might be influencing the outcomes, and identifying these specific factors could be challenging. Additionally, because the data were obtained from a subset of nurses working in radiology departments in Japan, there may have been some bias in the sample. However, we believe that the information obtained from nurses involved in radiology practice is valuable, and that it can serve as an opportunity to consider improvements in the work environment and support systems for nurses. Moreover, this study did not pilot test using a sociodemographic questionnaire, and it was not possible to determine the nature of each affiliated institution or region. In addition, this study used a self-administered questionnaire, therefore, factors that were not observed may still exist. Additionally, the questionnaire was designed to assess the subjects' perceptions and did not objectively evaluate aspects such as workplace systems. Therefore, the possibility of a lack of objective assessment cannot be denied.

## CONCLUSIONS

This study was conducted to identify factors associated with work engagement among nurses working in the radiology departments of prefectural designated cancer care hospitals in Japan. The results revealed that age, gender, position, desire to be assigned to the radiology department, and having a consultation system for radiation exposure in the workplace were significantly associated with work engagement. Regarding having a consultation system for radiation exposure in the workplace, the results suggest that highly specialized nurses such as CNSs and/or preceptorship may play a role in reducing radiation exposure anxiety among nurses working in radiology departments, thereby increasing work engagement.

## ACKNOWLEDGEMENTS

We would like to thank all the participants. We thank Benjamin Knight, MSc., from Edanz for editing a draft of this manuscript.

### Funding

The authors received no funding for this work.

### Competing Interests

The authors declare that they have no competing interests.

### Author Contributions

- Hitomi Tsuchihashi conceived and designed the experiments, performed the experiments, analyzed the data, prepared figures and/or tables, and approved the final draft.

PeerJ

- Takumi Yamaguchi conceived and designed the experiments, analyzed the data, prepared figures and/or tables, and approved the final draft.
- Yumiko Yamada performed the experiments, authored or reviewed drafts of the article, and approved the final draft.
- Tamami Koyama performed the experiments, authored or reviewed drafts of the article, and approved the final draft.
- Yuko Matsunari performed the experiments, authored or reviewed drafts of the article, and approved the final draft.

## Human Ethics

The following information was supplied relating to ethical approvals (*i.e.*, approving body and any reference numbers):

The present study was conducted in accordance with the tenets of the Declaration of Helsinki and approved by the Ethics Committee of Kagoshima University Epidemiological Study (approval number: 210184).

## Data Availability

The raw measurements are available in the Supplemental File.

## Supplemental Information

Supplemental information for this article can be found online at http://dx.doi.org/10.7717/peerj.18426#supplemental-information.

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
