# Peer review of "Factors associated with work engagement of nurses in the radiology department, Japan: a cross-sectional study"

_PeerJ, doi:10.7717/peerj.18426_

## Round 0.1 · original submission · Minor Revisions

The reviewers only have minor comments which you should address. Please incorporate their comments in an appropriate revision.

·

Basic reporting

Thank you so much for the invitation to review this article; Kindly consider:
1)Abstract: Background differs from aim, even; though the mentioned is an unclarified aim. kindly consider the aim as the title and mention the background and brief paragraph of the literature review. Methods (mention the type of research design, sampling type, and setting); Results (consider mentioning part of descriptive data with percentage); Keywords (where???) kindly add it and sort alphabetically.
2)Introduction: Clear, complete but there are more references in the texts
3)Materials & Methods: Setting (mention the setting here); Participants (mention the subject only, type of sampling, and how to estimate); consider the tools under (Tools of data collection)
4)Ethical Consideration: Mention the date of approval
5)Results (consider commenting on all tables and figures and numbering them); table 1 has a technical issue needs to revise the subcategory of each item.
6)discussion (clear, discussed and complete)

Experimental design

No comment

Validity of the findings

No comment

Reviewer 2 ·

Basic reporting

General comments
The title of the article "Factors associated with work engagement of nurses in a radiology department, Japan: A cross-sectional study" is very interesting and refers to an interesting topic. The title properly explains the purpose and objective of the article.
The abstract of the article is structured and contains the necessary parts: background, discussion methods, results and keywords. The abstract contains a proper summary of the article, the language used in the abstract is easy to read. The authors provide appropriate background on the topic and rationale for this article and describe what the authors hope to achieve.

In the introduction, the authors clearly and concisely describe the factors associated with the work engagement of radiology nurses in Japan. The research design is described in detail.

The research design is adequate and does not contain any particular deficiencies.
The measuring instrument is clearly described.
The population of interest and the sampling procedure are clearly defined.
The data collection procedure is clearly described.
Results: the results are clearly presented, the authors provide accurate research results, and there is sufficient evidence for each result.

I suggest that the authors write table 1 more clearly. The authors entered only n (%) in the table, not the standard deviation. However, in the notes at the bottom of the table, the authors wrote the standard deviation.
I suggest that the authors write tables 2 and 3 more clearly. The authors entered the mean value (SD) in the table. I suggest that the authors write Notes: SD=standard deviation.

The conclusion is clear.
Finally, this was interesting information about the article. In its current state, it adds many new insights.


Basic reporting
The article is written in professional English. ).
Literature, sufficient background/context of the field.
The article contains enough introduction and background to show how the work fits into the wider body of knowledge. Relevant previous literature is cited appropriately.

Experimental design

Research question well defined, relevant and meaningful. It was stated that the research fills a perceived gap in knowledge about factors associated with work engagement of nurses in the radiology department.
The research question is a clear question. The authors identified factors associated with nurses' work engagement.

A rigorous investigation conducted in accordance with high technical and ethical standards.
The research was conducted in accordance with the applicable ethical standards in the field.

Methods described with sufficient detail and information for replication.
The methods are described with enough information that another examiner can repeat them.

Validity of the findings

All underly data have been provided.

---

## Round 0.2 · accepted · Accept

Thank you for addressing the reviewer feedback when revising your paper. I think this helped improve the clarity in some important places and the paper is now ready for publication.

Reviewer 2 ·

Basic reporting

The authors accepted the reviewers' suggestions and made changes.

Experimental design

The authors accepted the reviewers' suggestions and made changes.

Validity of the findings

The authors accepted the reviewers' suggestions and made changes.

Additional comments

The authors accepted the reviewers' suggestions and made changes.